# Factors Affecting Disaster or Emergency Coping Skills in People with Intellectual Disabilities

**DOI:** 10.3390/bs13121018

**Published:** 2023-12-18

**Authors:** Eun-Young Park

**Affiliations:** Department of Secondary Special Education, Jeonju University, Jeonju 55069, Republic of Korea; eunyoung@jj.ac.kr; Tel.: +82-63-220-3186

**Keywords:** people with intellectual disabilities, disaster, emergency, coping skill, education level, training

## Abstract

This study aimed to investigate the disaster or emergency coping skills of people with intellectual disabilities and the factors that affect these skills. The panel survey on the lives of people with disabilities from the 3rd dataset (2020) of the Korea Development Institute for the Disabled was used for this analysis. Response data from 275 people with intellectual disabilities aged 10 years or older were analyzed. Differences between disaster or emergency coping skill levels and sub-questions of skills, according to the general characteristics of people with intellectual disabilities, were identified, as well as factors affecting the level of disaster or emergency coping skills. The results show that the coping skills level was low; among the sub-questions, the use of fire extinguishers and awareness of the location of fire extinguishers or emergency bells in the event of a disaster or emergency were also low. Factors affecting the level of coping skills were found to be the level of education and experience in comprehensive disaster coping training. The results of this study suggest that training and education on disaster or emergency coping skills for people with intellectual disabilities are necessary and that programs should be developed for this purpose.

## 1. Introduction

Human lives are threatened daily by the rapid growth of society and the increase in various social crises, such as large industrial accidents. At the 3rd UN World Conference on Disaster Risk Reduction in 2015, the Sendai Framework was adopted as a follow-up to the existing Hyogo Framework for Action, with promising action guidelines for each country to reduce disasters [1]. The goal of disaster risk reduction in the Sendai Framework emphasizes risk management for disaster-vulnerable groups, which is important for contemporary sustainable development [2]. The purpose of the Hyeogo Framework is to develop and implement strategies to promote disaster risk reduction and response at the international level, while the Sendai Framework aims to promote multi-stakeholder participation in disaster management, strengthening disaster tolerance, and promoting understanding.

In a 2013 UN survey, approximately 20% of more than 5000 people with disabilities in 126 countries responded that they were able to evacuate immediately without difficulty in the event of a sudden disaster or emergency situation [3]. Disaster incidents have garnered a global focus on the challenges faced by individuals with disabilities during disasters. In the aftermath of Hurricane Katrina in 2005, elderly residents found themselves trapped in St. Rita’s Nursing Home, some in wheelchairs and beds, as floodwaters surged, resulting in tragic drownings. Similarly, during the Great East Japan Earthquake and Tsunami in 2011, the mortality rate of people with disabilities was more than twice as high as that of the general population [4]. 

People with disabilities are among the most vulnerable groups and, therefore, at a high risk of suffering injury due to their existing disabilities and behavioral problems [5,6]. The World Report on Disability highlights the limited general research conducted on disability and underscores the urgent necessity for data-driven studies to guide policy decisions and address issues pertaining to the experiences of people with intellectual disabilities in disaster situations [7]. 

Considering the vulnerability of people with intellectual disabilities to disasters, it can be said that it is especially important for disaster-vulnerable groups to have the ability to respond appropriately to disasters before they occur. Kett and van Ommeren [8] suggested that individuals with cognitive disabilities should be prioritized in war and conflict scenarios because of their increased risk of abuse and premature death. Given that cognitive disability is a major concern, people with intellectual disabilities are likely to require assistance to respond effectively to emergency situations. Research on the impact of intellectual disabilities in disaster and conflict situations is scarcer than research on other disability types [7]. 

An analysis of data derived from participants in the 2006–2007 Behavioral Risk Factor Surveillance System survey [9] revealed that only 25.8% of individuals with disabilities felt “highly prepared” for an emergency, whereas 20.7% indicated that they were not prepared. A study examining the coping strategies of 150 individuals with disabilities during natural disasters in the coastal areas of Bangladesh, particularly in the Mongla, Rampal, and Sharankhola regions of the Bagerhat district, reported that a significant proportion of respondents (60%) did not receive disaster preparedness training. However, the majority (88%) were aware of the nearest government-designated disaster shelters and sought refuge there before or during the disaster [10].

Disaster and emergency coping skills are essential for people with intellectual disabilities to live independently in the community. To effectively educate people on disasters and emergency coping skills, it is necessary to first determine their ability or level of these skills. In this study, we use panel data on the lives of people with disabilities to verify the level of disaster or emergency coping skills of people with intellectual disabilities and the variables that affect the level of their coping skills. If the level of disaster and safety coping skills and related variables of people with intellectual disabilities are revealed through this study, it is expected to help provide basic information necessary to determine how vulnerable people with intellectual disabilities are and to develop safety education and training programs.

## 2. Materials and Methods

### 2.1. Data

This study used the 3rd dataset (2020) of the Korea Development Institute for the Disabled’s panel survey on the lives of people with disabilities. The data consists of the first year of 2018 data, the second year of 2019, and the third year of 2020. The survey participants were persons classified as disabled by the Welfare of Persons with Disabilities Act. Among the disabled people registered between 1 January 2015 and 31 December 2017 with the Ministry of Health and Welfare, 6121 persons with disabilities and their household members participated in the panel survey. Considering disability type, degree of disability, sex, etc., the number of survey subjects was determined. The survey questions of the panel consisted of disability acceptance, health and medical care, independence, and social participation. Among the types of disabilities, there were 352 people with intellectual disabilities. In this study, data from 275 people with intellectual disabilities aged 10 years or older were used for analysis, and the remaining 77 data, which were ages below 10 were not used for analysis. The research method for the Panel Survey on the Lives of People with Disabilities was a face-to-face, one-on-one interview conducted by a professional interviewer using a tablet PC as the survey tool (TAPI: Tablet-Assisted Personal Interviewing).

### 2.2. Measure

#### 2.2.1. Independent Variable

Factors affecting the disaster or emergency coping skills of people with intellectual disabilities consist of general characteristics such as gender, educational status, age, and experience in comprehensive disaster response training. Gender was measured as 1 for male and 2 for female, and educational status was measured as 0 for no education, 1 for elementary school, 2 for middle school, 3 for high school, 4 for college, and 5 for university education. Age was categorized as follows: 1 for teenagers, 2 for those in their 20s, 3 for those in their 30s, 4 for those in their 40s, 5 for those in their 50s, 6 for those in their 60s, 7 for those in their 70s, and 8 for those 70 years of age and above. Experience of comprehensive disaster coping training was measured as 1 if experienced and 2 otherwise.

#### 2.2.2. Dependent Variable

The dependent variables were disaster or emergency coping skills. A common definition of disaster or emergency coping skills is the concepts of prevention, preparedness, response, and recovery [11]. Disaster or emergency coping skills are defined as a systematic control activity to recognize and respond in advance to the risks of incidents and accidents that cause damage to humans, to prevent disasters from occurring in advance and to manage risk factors and damage effectively when they occur [12]. The scale for measuring disaster or emergency coping skills was created by panel researchers based on the lives of persons with disabilities, referring to the definition of disaster management presented in previous studies, crisis situation training guides, and manuals for each type of disaster [11,12,13,14]. Disaster or emergency coping skills consist of six questions about how well one is aware of the response measures to be taken in the event of a crisis and are rated on a 4-point Likert scale (1 = not at all, 2 = hardly at all, 3 = somewhat able, 4 = quite able). The specific questions were: (1) In the event of a disaster or emergency, you are able to report it to the fire department or police station. (2) You can notify others about disasters or emergencies. (3) You know where to locate the fire extinguishers and emergency bells in the house/building. (4) You know how to use fire extinguishers. (5) In the event of a disaster or emergency, you can move to a shelter. (6) You can recognize disasters or emergency situations. In this study, Cronbach α = 0.927 (0.912~0.939).

#### 2.2.3. Statistical Analysis

Descriptive statistics were used to determine the general characteristics of the study participants. An independent t-test and a one-way analysis of variance were performed to determine the differences in disaster response skills according to related variables. The Spearman correlation test was employed to test for multicollinearity among the variables, which was not observed in this study. Additionally, the variance inflation factor (VIF) of each independent variable was checked. The VIF ranged from 1.039 (sex) to 1.153 (experience in comprehensive disaster response training), and there was no multicollinearity. Individual IDs with sex, age, size of residential area, presence of comorbid disabilities, and experience in comprehensive disaster response training served as independent variables. A stepwise selection was applied during the multiple regression analysis to determine the influence of these variables on disaster response skills. In case of missing values, the option to exclude them from the list was selected and analyzed. The statistical significance level was set as α = 0.05.

## 3. Results

### 3.1. General Characteristics

Table 1 shows the general characteristics of people with intellectual disabilities and their level of disaster response skills according to general characteristics. The gender ratio was 147 men (53.5%) and 128 women (46.5%), and there was no difference in disaster response skills by gender. In terms of age, the 20s age group was the largest, with 90 people (32.73%), and with this group, there appeared to be differences in disaster response skills by age. People in their 30s were found to have the best disaster response skills (M = 3.07, SD = 0.84), whereas those in their 70s and older were found to have the lowest disaster response skills (M = 2.04, SD = 0.48). In terms of education level, most participants had graduated from high school (138, 50.18%). There was a significant difference in disaster response skills according to the education level. Those who graduated from junior college had the highest level of disaster or safety coping skills (M = 3.19, SD = 0.70), followed by those who graduated from high school (M = 2.87, SD = 0.76). Most people with intellectual disabilities had no comorbid disabilities, and in terms of the size of their area of residence, most lived in cities.

### 3.2. Level of Disaster or Emergency Coping Skills

Table 2 shows the level of disaster or emergency coping skills across the sub-questions. The lowest level was calculated for the ability to use fire extinguishers (M = 2.15, SD = 1.016). The second lowest level indicated whether they knew where to locate the fire extinguishers and emergency bells in the house/building (M = 2.60, SD = 1.007). The highest level was calculated for the ability to notify others about disasters or emergencies (M = 3.03, SD = 0.918).

### 3.3. Factors Affecting the Disaster or Safety Coping Skill

Table 3 presents the regression analysis results. Education level and experience in comprehensive disaster-response training had a significant effect on the level of disaster-response skills. The standardized coefficients were 0.191 for education level and −0.190 for experience with comprehensive disaster response training. The explanatory power was 12.9%.

## 4. Discussion

People with disabilities are vulnerable to disasters, and there are safety concerns to consider [15]. Among the vulnerable disaster-safety groups, people with disabilities suffer great harm because their ability to recognize, judge, and respond to disaster situations is not the same as that of non-disabled people. People with disabilities are likely to be at greater risk of self-injury or harm from potentially dangerous situations due to deficits in critical skills for navigating situations, such as social, motor, and communication skills [16,17]. The results of this study confirmed that people with intellectual disabilities are a vulnerable group in coping with disasters and emergencies. The average level of disaster or emergency coping skills according to age was 3.00 for those in their 20s and 3.07 for those in their 30s, indicating that they could cope with a disaster or emergency, while all other age groups were found to be hardly able to do so. The level of disaster or emergency coping skills according to educational background was found to be poor for all except college graduates.

When divided into sub-questions, the question with the lowest level among the sub-questions of disaster or emergency coping skills was the ability to use fire extinguishers. The average knowledge of how to use a fire extinguisher was 2.15, followed by the awareness of the location of fire extinguishers and emergency bells within the house/building, which was the lowest, at 2.60. The incidence of injuries among people with intellectual disabilities is reportedly higher than that in the general population [18], and the probability of dying from a fire is four times higher than that of the general population [19]. The leading causes of accidents in people with intellectual disabilities are falls, suffocation, and drowning, most of which occur at home, unlike those in the general population [6]. Considering that people with intellectual disabilities have a high probability of experiencing an accident at home, it is essential to be aware of the location of emergency bells in the house.

The variables affecting the level of disaster or emergency coping skills of people with intellectual disabilities were the level of education and comprehensive disaster response training. Despite the importance of education, people with disabilities have little educational experience regarding action tips in the event of a disaster or emergency accident [20]. In this study, only 13.5% (37 people) of the people with intellectual disabilities had experience in comprehensive disaster response education and training, indicating that the proportion with training experience was very low. Although the frequency of disasters is low, a single disaster can cause enormous damage to the economy and human lives, making repeated education and training important [21]. It is crucial for people with intellectual disabilities to develop the skills to recognize risky situations and respond appropriately, as relying solely on other persons’ protection or preventive measures may not be sufficient. Disaster or safety coping skills through education becomes paramount. Education programs for disaster or emergency coping skills include understanding potential dangers, assessing the situation, and taking appropriate actions to ensure personal safety. In particular, the results of this study show that, to improve the disaster response skills of people with intellectual disabilities, disaster response education and information provision plans that target those with low levels of education need to be strengthened.

An important goal for persons with severe disabilities is to enhance their independence and ultimately integrate into the community environment [22]. The traditional approach to disaster preparedness focuses on the vulnerability of children with disabilities and views them as dependent on and in need of care [23]. However, recent studies suggest that although it is necessary to prepare families and teachers for children with disabilities in disasters or dangerous situations, the children themselves should also be further empowered [24,25]. Research has shown that even children with disabilities can prepare for, respond to, and recover from dangerous situations [26]. People with intellectual disabilities require special education on the risk and safety response [27] and can achieve results through special education. In a survey of the post-training evacuation behavior of people with intellectual disabilities in residential facilities in Ireland, 80% of the participants were able to evacuate within 8 min during nighttime training [28]. Evidence of effective interventions for safety skill training has been consistently reported in people with intellectual disabilities. Representative interventions include video modeling, facilitation, and situational training. Ozkan [29] provided training on first aid skills to three children with intellectual disabilities using peer and self-video modeling and reported the effect of the video modeling method for improving safety skills. Similarly, Bassette et al. [30] taught students with intellectual disabilities how to use cell phones as a safety technique using a video modeling intervention with minimal facilitation, and all participating students learned the task and communicated their place in the community through sending a photo. Behavioral skill training has been used consistently and has been implemented to target a variety of dependent variables, such as kidnapping prevention skills [31] and pedestrian skills [32]. For example, Knudson et al. [33] implemented behavioral skill training to teach people with intellectual disabilities how to evacuate their residence after hearing a fire alarm.

This study is in context with a previous study that identified the lower level of disaster or safety coping skills of people with intellectual disabilities. Moreover, this study includes meaningful variables related to the level of disaster or safety coping skills through a systematic sampling panel data analysis. Given that previous research indicates lower awareness and coping skills for disaster preparedness among people with intellectual disabilities, the main contribution of this study was that the level of skills according to general characteristics and education level and the experience of comprehensive disaster response training was verified as influencing skill level.

It is, however, important to highlight the limitations of this study. First, it was not possible to investigate the various variables that could affect the level of disaster or safety-coping skills. The level of disaster or safety-coping skills is influenced by the level of knowledge of disasters or safety. The panel data did not provide information on the level of knowledge, so it could not be analyzed, but it is believed that future research on the level of knowledge and technology will be necessary. Second, although experience in comprehensive disaster-response training was found to affect disaster and safety response skills, information on the impact of the content, type of intervention, intensity, and frequency of training was not confirmed. Information on the content, intervention type, frequency, and intensity of training must be provided to design an effective training program. Previous studies have explored the efficacy of teaching safety skills within real or simulated scenarios to instill effective disaster or safety coping skills [34]. In these instances, practical applications in situations where risk is likely to manifest have proven beneficial. Moreover, for the design of an effective training program, a comprehensive understanding of the content, frequency, and intensity of training is essential. This information ensures that the training curriculum aligns with the specific needs and challenges of the target audience. Tailoring training programs to address the unique requirements of different groups or contexts enhances the relevance and applicability of the acquired skills. While the positive impact of comprehensive disaster-response training on disaster and safety response skills is acknowledged, further exploration into the nuanced factors of intensity and frequency is warranted.

## 5. Conclusions

This study confirmed that the level of disaster or safety coping skills of people with intellectual disabilities is low and that the level of coping skills is influenced by the level of education and experience of comprehensive disaster response training. The results of this study suggest that efforts are needed to improve the disaster or safety-coping skills of people with intellectual disabilities. Particularly, it is necessary to include training to be cognizant of the location of the fire extinguishers and emergency bells within a building or house in the event of a disaster or safety situation. Education level and experience in comprehensive disaster response training were identified as variables affecting the level of disaster and safety response skills of people with intellectual disabilities. These results show that in addition to efforts to improve the education level of people with intellectual disabilities, specific programs targeting those with low levels of education need to be developed. This suggests a necessity to implement disaster training programs for people with intellectual disabilities on a continuous basis.

## Figures and Tables

**Table 1 behavsci-13-01018-t001:** General characteristics and disaster or safety coping skills level of participants.

Variables	*n*	%	*X* ^2^	M	SD	*t*/F
Sex						
Male	147	53.5	1.313	2.86	0.83	0.352
Female	128	46.5		2.65	0.79	
Age						
10–19	51	18.55	102.160 **	2.78	0.94	5.981 **
20–29	90	32.73		3.00	0.75	
30–39	32	11.64		3.07	0.84	
40–49	36	13.09		2.63	0.74	
50–59	34	12.36		2.46	0.62	
60–69	24	8.73		2.28	0.75	
70+	8	2.91		2.04	0.48	
Education level						
Elementary school graduate	42	15.27	257.080 **	2.35	0.76	4.929 **
Middle school graduate	52	18.91		2.63	0.83	
High school graduate	138	50.18		2.87	0.76	
College graduate	27	9.82		3.19	0.70	
University graduate	11	4.00		2.55	1.10	
No education	5	1.82		2.53	1.12	
Comorbid disability						
Yes	24	8.7	187.378 **	2.58	0.72	1.972
No	251	91.3		2.78	0.83	
Residential area						
Big city	115	41.8	174.909 **	2.68	0.85	0.990
Small and medium-sized city	136	49.5		2.82	0.77	
Rural	24	8.7		2.78	0.89	

Note: M = mean, SD = standard deviation, ** *p* < 0.001.

**Table 2 behavsci-13-01018-t002:** The level of disaster or emergency coping skills.

Variables	M	SD	Mean Rank	*X^2^*	*p*
In the event of a disaster or emergency, you are able to report it to the fire department or police station ^acd^.	2.84	0.929	3.66		
You can notify others about disasters or emergencies ^b^.	3.03	0.918	4.09	363.351	<0.001
You know where to locate the fire extinguishers and emergency bells in the house/building ^d^,	2.60	1.007	3.19		
You know how to use fire extinguishers ^a^.	2.15	1.016	2.27		
In the event of a disaster or emergency, you can move to a shelter ^d^.	2.94	0.974	3.80		
You can recognize disasters or emergency situations ^c^.	3.00	0.892	3.99		

Note: M = mean, SD = standard deviation, Items with different superscript letters within moderators are significantly different at the *p* < 0.05 level.

**Table 3 behavsci-13-01018-t003:** Summary of multiple regression analysis for the level of disaster or safety coping skills.

Variable	B	SE	*β*	*t*	*p*
Sex	−0.216	0.124	−0.157	−1.736	0.085
Child’s age	−0.064	0.047	−0.127	−1.355	0.178
**Education level**	**0.139**	**0.066**	**0.191**	**2.111**	**0.037**
Presence of comorbid disability	0.180	0.222	0.073	0.811	0.419
Size of residential area	0.000	0.100	0.000	0.003	0.998
**Experience in disaster response comprehensive training**	**−0.281**	**0.141**	**−0.190**	**−1.993**	**0.049**
*R* ^2^			0.129		

Note: B = unstandardized coefficient; SE = standard error; *β* = standardized coefficient; Bold means significance difference at *p* = 0.05 level.

## Data Availability

Data can be requested at https://www.koddi.or.kr/intro/dataOpenInfo.jsp accessed on 10 December 2021. The data provided in this study are public data because they are panel survey data provided by Korea’s Disabled People’s Development Institute. If you go to the site presented above, you will see a tab named ‘Public Data Request,’ and you can receive the data by clicking the tab and requesting the necessary data.

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
