# Peer review of "Factors Affecting Disaster or Emergency Coping Skills in People with Intellectual Disabilities"

_behavsci, 2023, doi:10.3390/bs13121018_

Round 1
Reviewer 1 Report
Comments and Suggestions for Authors
Thank you for sharing this important and timely topic about the coping skills of PWID. This is an important area to consider to develop strategies and training to assist PWID during disasters. A few suggestions to improve the paper are given below:
1. Please define PWID to be more specific as there are different kind of disabilities.
2. Define coping/coping skills and how it is perceived among PWID would be helpful. Additional literature around coping among PWID would be helpful in the paper to gather a clear understanding of the topic.
2. Method is written well. Please specify which data was used and provide the citation of the database or more details about the database used.
3. Additional recommendations to improve disaster training specifically in the context of PWID is encouraged. You said special education, it would be advisable if you can elaborate more. Also, how disaster providers can help PWID during disaster situations or how they should be trained can also be added as recommendation to further strengthen the paper.
Comments on the Quality of English LanguageMinor editing is required.
Author Response
I have addressed all your suggested edits and comments. I hope that I have adequately strengthened my manuscript.
Comment #1. Please define PWID to be more specific as there are different kind of disabilities.
Response #1: We worked together to correct all abbreviations (People with intellectual disabilities, Throughout manuscript).
Comment #2: Define coping/coping skills and how it is perceived among PWID would be helpful. Additional literature around coping among PWID would be helpful in the paper to gather a clear understanding of the topic.
Response #2: Additional literature around coping skills have been inserted (line 111 ~116).
Commonly, Disaster or emergency coping skills refers to the concept of prevention, preparedness, response, and recovery [11]. It is defined as a systematic control activity to recognize and respond in advance to the risks of incidents and accidents that cause damage to humans, and to prevent disasters from occurring in advance and manage risk factors and damage effectively when they occur [12].
Comment #3: Method is written well. Please specify which data was used and provide the citation of the database or more details about the database used.
Response #3: Details have been inserted in Data availability (line 305 ~ 309).
Data Availability Statement: Data can be requested at https://www.koddi.or.kr/intro/dataOpenInfo.jsp. The data provided in this study are public data because they are panel survey data provided by the Korea’s Disabled People’s Development Institute. If you go to the site mentioned above, you will see a tab named ‘Public Data Request’, and you can obtain the data by clicking the tab and requesting for it.
Comment #4: Additional recommendations to improve disaster training specifically in the context of PWID is encouraged. You said special education, it would be advisable if you can elaborate more. Also, how disaster providers can help PWID during disaster situations or how they should be trained can also be added as recommendation to further strengthen the paper.
Response #4: Additional recommendations have been inserted in discuss section (line 242 ~ 254).
Evidence of the effective interventions for safety skill training has been consistently reported by people with intellectual disabilities. Representative interventions include video modelling, facilitation, and situational training. Ozkan [29] provided training on first aid skills to three children with intellectual disabilities using peer and self-video modelling, and reported the effects of video modelling method for improving safety skills. Similarly, Bassette et al. [30] taught students with intellectual disabilities how to use cell phones as a safety technique using a video modeling intervention with minimal facilitation, and all participating students learned the task and communicated their place in the community through sending phot. Behavioural Skill Training has been consistently used and implemented to target a variety of dependent variables, such as kidnapping prevention skills [31] and pedestrian skills [32]. For example, Knudson et al. [33] implemented behavioural skill training to teach people with intellectual disabilities how to evacuate their residence after hearing a fire alarm.
Reviewer 2 Report
Comments and Suggestions for Authors
Please see attached.

Author Response
I have addressed all your suggested edits and comments. I hope that I have adequately strengthened my manuscript.
Introduction
Comment #1: Please provide additional context information about the Sendai Framework and Hyogo Framework. Without this information, readers may not comprehend the objectives and scope of these two frameworks.
Response #1: More detail description about Frameworks have been inserted (line 32 ~ line 36).
The purpose of the Hyeogo Framework is to develop and implement strategies to promote disaster risk reduction and response at the international level, while the Sendai Framework aims to promote multi-stakeholder participation in disaster management, strengthening disaster tolerance, and promoting understanding.
Comment 2: On line 56, where the paragraph begins with "Research on the disaster response," I think the paragraph is the literature review covering previous studies related to the topic. Currently, the writing seems more descriptive than analytical, merely describing each study. I recommend analyzing each study and summarizing the key takeaways for readers. Clarify what you want readers to learn from these previous studies, identify any existing gaps, and consider expanding the literature review by adding more relevant studies.
Response #2: The first sentence was deleted because it seemed to cause confusion. After this sentence, the contents of each document are summarized and presented (line 60 ~ line 68).
An analysis of data derived from the 2006-2007 Behavioral Risk Factor Surveillance System survey [9] revealed that only 25.8% of individuals with disabilities felt "highly prepared" for an emergency, whereas 20.7% indicated that they were not prepared. A study examining the coping strategies of 150 individuals with disabilities during natural disasters in the coastal areas of Bangladesh, particularly in the Mongla, Rampal, and Sharankhola regions of the Bagerhat district, reported that a significant proportion of respondents (60%) did not receive disaster preparedness training. However, the majority (88%) were aware of the nearest government-designated disaster shelters and sought refuge there before or during the disaster [10].
Method
Comment #1: Section 2.1 serves as the introduction to the panel survey. I recommend that the author rewrite this section to improve clarity. As the data were gathered through the survey, there's a need for more comprehensive information that is currently missing, such as details on the distribution method of the survey and whether it was self-reported. Additionally, it would be beneficial to clarify whether people with disabilities took the survey independently or with assistance from others.
Response #1: Related information about panel data have been inserted (line 82 ~ line 97).
This study used the 3rd dataset (2020) of the Korea Development Institute for the Disabled's panel survey on the lives of people with disabilities. The data consists of the first-year of 2018 data, the second year of 2019, and the third year of 2020. The survey participants were individuals classified as disabled by the Welfare of Persons with Disabilities Act. Among the disabled people registered between January 1, 2015 and December 31, 2017 with the Ministry of Health and Welfare, 6,121 persons with disabilities and their household members participated in the panel survey. Considering disability type, degree of disability, sex, etc., the number of survey subjects was determined. The survey comprised questions on disability acceptance, health and medical care, independence, and social participation. There were 352 people with intellectual disabilities. In this study, data from 275 people with intellectual disabilities aged 10 years or older were used for analysis and the data of remaining 77 who were below 10 years were not used for analysis. The research method for the Panel Survey on the Lives of People with Disabilities involved a face-to-face, one-on-one interview conducted by a professional interviewer using a tablet PC as the survey tool (TAPI: Tablet-Assisted Personal Interviewing).
Comment #2: Section 2.1 contains many confusing elements, like "3rd dataset," which lacks clear explanation. It would be helpful for the author to provide more context to ensure a better understanding of this term.
Response #2: More detail information has been added (line 83 ~ line 84).
The data consists of the first-year of 2018 data, the second year of 2019, and the third year of 2020.
Comment #3: Regarding the sentence “Among the disabled people registered between January 1, 2015 and December 31, 2017 with the Ministry of Health and Welfare, 6,121 persons with disabilities and their household members formed a panel,” I do not understand it. Did 6121 persons with disabilities and their household members fill out the survey? What does it mean by forming a panel?
Response #3: That sentence has been revised (line 86 ~ line 98).
. Among the disabled people registered between January 1, 2015 and December 31, 2017 with the Ministry of Health and Welfare, 6,121 persons with disabilities and their household members participated in the panel survey.
Comment #4: Regarding the sentence “The number of survey subjects was determined according to the results of the sample distribution and sampling was done considering disability type, degree of disability, sex, etc., according to the sample design,” what does it mean?
Response #4: That sentence has been revised (line 88 ~ line 91).
Considering disability type, degree of disability, sex, etc., the number of survey subjects was determined. The survey comprised questions on disability acceptance, health and medical care, independence, and social participation.
Comment #5: Regarding the sentence “The panel survey on the lives of people with disabilities included; disability acceptance and change, health and medical care, independence, and social participation,” what does it mean?
Response #5: That sentence has been revised (line 88 ~ line 91).
Considering disability type, degree of disability, sex, etc., the number of survey subjects was determined. The survey comprised questions on disability acceptance, health and medical care, independence, and social participation.
Comment #6: Regarding the sentence “Among the types of disabilities, there were 352 PWID; in this study, data from 275 PWID aged 10 years or older were used for analysis,” the author did not use 352 data because of the age? Please clarify.
Response #6: That sentence has been revised (line 91 ~ line 93).
There were 352 people with intellectual disabilities. In this study, data from 275 people with intellectual disabilities aged 10 years or older were used for analysis and the data of remaining 77 who were below 10 years were not used for analysis.
Comment #7: Section 2.2.1 introduced independent variables, but in the data analysis and results sections, two additional independent variables, "Presence of comorbid disability"
and "Size of residential area," were presented. Are these not considered independent variables?
Response #7: This is because the number of cases with comorbid disability and the number of cases living outside the city (rural areas) were relatively small.
Discussion
Comment #1: What is the primary contribution of this study? Given that previous research indicates lower awareness and coping skills for disaster preparedness among people with disabilities, what is the main contribution of this study?
Response #1: Main contribution have been added (line 259 ~ line 263).
Given that previous research indicates lower awareness and coping skills for disaster preparedness among people with intellectual disabilities, the main contribution of this study was that detail level of skills according to general characteristics and education level and experience of comprehensive disaster response training have verified affected factors on skill level.
Comment #2: How does this study align with or differ from previous studies?
Response #2: Similarity and differences between this study and previous studies have been added (line 256 ~ line 259).
This study is in context with a previous study [4-7] identifying the lower level of disaster or safety coping skills of people with intellectual disabilities. And this study used meaningful variables related to the level of disaster or safety coping skills through a systematic sampling panel data analysis.
Comment #3: I believe this study adds value to the existing literature. I suggest the author discusses the implications for teaching disaster preparedness to people with intellectual disabilities. Particularly, considering that the identified key factors are education and training, the author can elaborate on how to provide effective training and education on disaster preparedness for people with intellectual disabilities. It would be beneficial to include more practical practices that practitioners can implement.
Response #3: More detail suggestion about program have been added (line 269 ~ line 282).
Second, although experience in comprehensive disaster-response training was found to affect disaster and safety response skills, information on the impact of the content, type of intervention, intensity and frequency of training was not confirmed, which is required to design an effective training program. Previous studies have explored the efficacy of teaching safety skills within real or simulated scenarios to instil effective disaster or safety coping skills [34]. In these instances, practical applications in situations where risk is likely to manifest have proven beneficial. Moreover, a comprehensive understanding of the content, frequency, and intensity of training ensures that the training curriculum aligns with the specific needs and challenges of the target audience and the training program is effective. Tailoring training programs to address the unique requirements of different groups or contexts enhances the relevance and applicability of the acquired skills. While the positive impact of comprehensive disaster-response training on disaster and safety response skills is acknowledged, further exploration into the nuanced factors of intensity and frequency is warranted.